# Review of Pediatric Tuberculosis in the Aftermath of COVID-19

Patrida Rangchaikul, Phillip Ahn, Michelle Nguyen, Vivian Zhong and Vishwanath Venketaraman *

College of Osteopathic Medicine of the Pacific, Western University of Health Sciences, Pomona, CA 91710, USA
* Correspondence: vvenketaraman@westernu.edu; Tel.: +1-909-706-3736; Fax: +1-909-469-5698

**Abstract:** In 2014, the World Health Organization developed the End Tuberculosis Strategy with the goal of a 95% reduction in deaths from tuberculosis (TB) by 2035. The start of the COVID-19 pandemic and global lockdown has had a major impact on TB awareness, screening, diagnosis, and prompt initiation of treatment, inevitably leading to a significant setback. We explore pediatric tuberculosis through the lens of the COVID-19 era, investigating how COVID-19 has impacted pediatric TB cases in different regions of the world and what the implications are for management moving forward to mitigate these effects. Furthermore, in light of recent findings showing how exposed infants and children are at higher risk than we thought of contracting the disease, greater attention and resources are needed to prevent further downward trends.

**Keywords:** tuberculosis; pediatrics; COVID-19; SARS-CoV-2; immunology; prevention

## 1. Introduction

Tuberculosis (TB) is an infectious bacterial disease caused by Mycobacterium tuberculosis (Mtb) that remains one of the leading causes of global childhood morbidity and mortality and the number one cause of death from infectious disease in children [1–4]. The New England Journal of Medicine's article in January 2022 reported that although COVID-19 has had devastating effects on every aspect of global health, TB services have been disproportionately affected with TB mortality rising worldwide for the first time in over a decade [2]. The WHO's Global Tuberculosis Report 2021 showed how "case notifications have plummeted due to pandemic-related disruptions in services" [5]. According to WHO estimations, 10 million people developed tuberculosis in 2020 but only 5.8 million cases were diagnosed and reported—an 18% decrease from 2019, mainly affecting Asian countries [2].

There are limited data and knowledge on how this translates to pediatric TB and how COVID-19 has impacted this particular subset of the population. Initial reports have shown a disproportionate increase in pediatric TB cases as well and has been possibly linked to the effects of stay-at-home measures causing an increase in household transmission to children [6]. According to the WHO, children represent around 10% of all TB cases [5]. In 2020, the WHO estimated that 1.1 million children became ill with TB worldwide with an average case fatality rate of 24% [4,5]. Low- and middle-income countries accounted for 98% of TB cases that were reported, and 30 countries with the highest TB burden accounted for 86% of new TB cases [5]. These countries are predominantly located in African and South-East Asian regions where 8 of the highest-burden countries accounted for two thirds of new TB cases [5].

In the US, rates of childhood tuberculosis have dropped at a similar rate to that seen in adult TB, falling from 2.1/100,000 in 1994 to 1.1/100,000 in 2007 [7]. In 2020, 315 cases of TB disease were detected among children ages 14 years or younger in the United States [8].

The incidence of TB disease and death among children was previously unknown due to a lack of emphasis on pediatric TB within most public health interventions [9]. Although TB is a curable and preventable disease, research on pediatric TB has become stagnant since

studies conducted prior to 1950 [10]. The ongoing disease burden among pediatric TB patients reveals the limitations of contemporary diagnostic measures and screening methods for children exposed to TB, even in the pre-pandemic era. Bacteriological confirmation is limited in the diagnosis of pediatric TB. Less than 15% of TB cases are sputum acid-fast bacilli smear-positive, and mycobacterial culture yields are 30–40% [11,12]. In countries where TB is not endemic, childhood TB is diagnosed mainly in symptomatic children. Screening is based on a triad of (1) close contact with an infectious patient, (2) a positive tuberculin skin test (TST), and (3) suggestive abnormalities on a chest radiograph [13]. However, case detection and contact tracing activities are not routine in TB programs of countries where TB is endemic [14,15]. As a result, the triad of criteria has limited application in these countries. In the past, transmission of pediatric TB has been largely monitored most effectively via household contact tracing [9]. However, in the pre-pandemic era, a systematic review conducted by Martinez et al. provided further evidence that TB transmission in high-burden areas most often occurs within the community rather than inside the household of a person who has TB [9].

Treatment methods for TB in children present another challenge in reducing the global TB burden. There is a limited number of pharmaceutical formulations that can be reliably used to treat TB in children. Other issues include a lack of information on the toxicity of drugs and unpredictable levels of adherence to treatment [16]. Preventive medications have shown to reduce the development of active TB disease by 91% among infected children if they are treated prior to experiencing TB symptoms [10]. However, it is difficult to deliver preventive therapy in a timely manner due to the challenges of investigating children who are suspected of having latent TB infection. Improvements in surveillance and screening methods are essential in reducing incidence rates in high-burden regions worldwide by detecting pediatric TB before it becomes an active disease [10]. Large-scale contact screening during the COVID-19 pandemic has become increasingly more important with recent estimations predicting an additional 6.3 million TB cases globally between 2020 and 2025 [17].

The high burden of childhood TB and poor outcomes are largely attributable to the challenges in confirming the diagnosis of pediatric TB and in screening for latent TB disease [17]. Globally, over 90% of children experience delays in receiving treatment since they are either never diagnosed or are misdiagnosed [18]. In 2019, more than 40% of households that were eligible for child contacts were not screened for tuberculosis disease or infection [18]. Therefore, expanding access to TB contact screening and providing preventive treatment for TB needs to be substantially improved. Additionally, COVID-19 has added on extra layers of complications and barriers to TB screening, as will be explored later on.

## 2. Pathogenesis of Childhood TB

Childhood TB symptoms are nonspecific, which poses a challenge to diagnosis, especially when confounded with the COVID-19 pandemic. For a quick overview of TB pathogenesis, TB is transmitted via the respiratory route when aerosolized small-droplet nuclei of an infected individual are inhaled into the alveoli of individuals in close contact [19]. There are numerous Mtb complex organisms, including M. africanum, M. bovis, and M. canetti that all act as causative agents of TB [20]. Once exposed to Mtb, children under the age of 5 years are at high risk of developing an infection that may then progress into active TB disease [21]. Due to age-related deficiencies or the down-regulation of important immunologic factors, infants and young children are the most susceptible to developing active and disseminated forms of TB, including TB meningitis [22,23].

However, according to a review conducted in 2017, the risk of TB disease follows a bimodal pattern, and children between the ages of 5 and 10 are at lower risk of progressive tuberculosis [19]. While children between the ages of 1 and 2 years have a 10–20% risk of TB pulmonary disease, children between the ages of 5 and 10 years have a 2% risk [19]. It is important to note that reemergence of Mycobacterium tuberculosis may occur among

adolescents who become more susceptible to active TB after several years of latency [24]. From ages 10 to 15 years, the risk of TB pulmonary disease in children returns to 10–20% [19].

The clinical presentation of pediatric TB has a wide variability, and patients often present with symptoms that are nonspecific [25–27]. Investigation of children who are suspected of having TB can be difficult, and identification of the bacterial pathogen is insufficient for confirming a diagnosis due to the paucibacillary nature of TB. A thorough clinical history, detailed physical examination, positive TB skin test, and imaging evaluation are all required to support and confirm a TB diagnosis [25–27].

The cardinal symptom in diagnosing TB is a persistent cough that is associated with radiologic and microbiologic evidence [28]. In clinical practice, pulmonary manifestation of TB is the most important issue in both pulmonary and extra-pulmonary TB infections. Other diagnostic symptoms include fever, fatigue, pallor, anorexia, failure to gain weight, and respiratory distress [29].

After exposure to Mtb, it requires approximately 4–12 weeks for adaptive immune responses to express any evidence of infection [30]. During this time, the infection extends into the lung parenchyma, which manifests clinically with nonspecific respiratory symptoms as well as hypersensitivity reactions such as fevers, erythema nodosum, or phlyctenular conjunctivitis [22]. At this time, chest radiographs also commonly reveal transient lymphadenopathy in the hilar and/or mediastinum regions [31,32]. Isolated intrathoracic lymphadenopathy is often not associated with symptoms. However, complications may occur as lymph nodes enlarge and inflammatory reactions continue to progress [19]. Approximately 6 months after exposure, these complications may manifest clinically with cough, dyspnea, malaise, chest pain, and fever. Approximately 1 to 2 years after primary infection, some children may develop calcifications that are associated with a reduced risk of further disease progression [30,31].

Once the mycobacteria are established within regional lymph nodes, it is possible for the infection to traverse via the lymphatics and into the systemic circulation. Hematogenous spread ensues and distal sites may become primed as a nidus for future complications [33,34]. In children with developing immune systems, this stage of TB progression confers a high risk of extrapulmonary TB. The peripheral lymph nodes and the central nervous system are the most common sites of extrapulmonary TB manifestations [19]. Lymphadenitis results in enlarged, painless lymph nodes in the cervical region that lacks warmth or erythema. As the disease progresses, it is possible that the nodes become more palpable, and sinus fistulas begin to form [19].

The most serious manifestation of extrapulmonary TB occurs when infection extends into the central nervous system (CNS). Complications within the central nervous system commonly present as TB meningitis and results in significant morbidity or mortality in approximately 50% of cases [35–37]. During the early stages of CNS infection, sub-acute and nonspecific symptoms emerge as systemic problems, including fever, anorexia, irritability, and focal respiratory or gastrointestinal symptoms. Diagnostic symptoms of meningitis become more apparent as the disease progresses and presents with meningismus, convulsions, vomiting, altered consciousness, cranial nerve palsies, or signs of intracranial pressure. Complications of CNS infection are represented by hydrocephalus, cerebrovascular disease, coma, and tuberculoma [13].

### 3. Immunology of Active TB Infection

For a better understanding of how the immune system in children at certain ages predisposes them to a greater risk of TB infection when exposed, we first review the immunology behind TB infection. Mtb is a facultative intracellular pathogen, infecting host cells of humans [38]. Mtb is commonly transferred from humans to humans but may also be transferred from animals to humans [39]. Mtb is transmitted by humans breathing in bacteria-laden aerosol droplets [40]. TB can manifest in many ways depending on the host's immune system. Mtb is unlike most pathogens as they do not have the classical

virulence factors to help invade a host cell [41]. Instead, they have developed and evolved to form a new infecting method that is more effective. The Ms pathogen can invade a host cell's granuloma and, specifically, the macrophages and dendritic cells, which are antigen-presenting cells (APCs) as well as lymphocytes [42]. Once the pathogen is inside the granuloma, it can avoid the immune response, replicate, and be transported to other sites in the body, causing infection in other organs as well.

There are two possible manifestations of TB, latent TB infection (LTBI) and active TB (ATB) [43]. LTBI manifests as asymptomatic, whereas ATB may display with different symptomologies. Individuals with LTBI maintain a reservoir of Mtb pathogens that perpetuates the continuation of TB infections, which is known as potential reactivation TB [44]. LTBI is defined as an individual who has Mtb antigens in their system, while not showing any clinical symptoms [45]. A systematic review shows that out of all TB infections, the percentage of ATB infection is about 3.1%, whereas the percentage of LTBI is about 51.5% [45]. The high proportion of TB infections that manifest as LTBI causes difficulty identifying all TB cases and preventing spread in a timely manner. Only about 5–10% of those who have LTBI will have worsened disease that turns into ATB [46]. The likelihood that a LTBI turns into symptomatic ATB is highest closest to the time of initial infection of Mtb [47].

The most common symptoms of TB affect the lungs and cause cough, fever, and weight loss. In pediatric patients, the most common manifestation of symptoms is pulmonary TB [45]. Children who have TB (either LTBI or ATB) comprise a large majority from which TB can arise.

Once Mtb enters APCs such as macrophages and dendritic cells, they prevent the host T cells from neutralizing and killing off the pathogen [48]. This bacterium enters the lung's alveolar macrophages initially, but these immune cells do not have a well-adapted mechanism to detect the pathogen, and therefore causes a delay in the immune response [48]. This delay in the activation of immune response causes the pathogen to replicate and cause disease. Mtb can effectively circumvent the host immune response by entering these alveolar macrophages and avoiding detection. Once this pathogen moves from the alveolar macrophages to the interstitial space, it then triggers the delayed adaptive immune response, and the host T cells can begin cell-mediated immune responses against the pathogen [48].

In a study looking at individuals with ATB, LTBI, healthy controls, and individuals who are cured of TB, it was found that there were significantly decreased levels of T lymphocytes in ATB individuals [49]. T lymphocytes are critical in the host immune response, producing cytokines such as interferon-gamma (IFN-g) and tumor necrosis factor alpha (TNF-a), which both induce uptake of the pathogen by macrophages and enhancement of antimycobacterial effector responses within the macrophages [50]. The macrophages can correctly identify Mtb by toll-like receptors (TLRs) and subsequently phagocytose the pathogen and control the infection [51]. The macrophages secrete other cytokines such as IL-12, which causes CD4 T cells to differentiate into the T-helper subset (Th1), which, in turn, secrete IL-2 and IFN-g [51]. As IFN-g and TNF-a have been found to be integral for the immune response against Mtb, a study found that immunizing mice with the TNF-a mutant caused increased CD4 and CD8 T cell responses, therefore neutralizing and killing off the bacteria faster and more efficiently [50]. There can also be a negative implication of too much TNF-a causing tissue and cell damage, as well as inflammation [50]. Having decreased levels of TNF-a have been found to correlate with a higher incidence of TB infection [52]. This means that these cytokines, including TNF-a, are critical in the immune response against TB infection.

T cells are crucial to the adaptive immune responses that combat Mtb infection. Both CD4 and CD8 T cells have been implicated in the immune response against Mtb infection. It was found that mice that did not have CD4 T cells died earlier after Mtb infection than mice without CD8 T cells [53]. This shows that CD4 T cells are more important in the host immune response in fighting off Mtb. Initially, when the Mtb pathogen enters the host, the

first immune response that becomes activated is the CD4 T cells [52]. Once the macrophages phagocytose the mycobacterium, it is transported to the mediastinal lymph nodes, and this is when CD4 T cell expansion begins [52]. It has been shown that dendritic cells that uptake the pathogen can release Mtb antigens that can be presented to the CD4 T cells and increase the adaptive immune response [52]. It has also been found that multi-functional T cells are essential in fighting off TB infection [54]. Multi-functional T cells can secrete more than one type of cytokine to increase the immune response against the pathogen. Having T cells that secrete more than one kind of cytokine allows multi-faceted effector responses leading to effective control of Mtb infection.

It is critical to quickly screen and identify any individuals with TB in order to prevent transmission and cease the spread of this bacteria. The tuberculin skin test (TST) and interferon-y release assays (IGRAs) are two ways to screen individuals for TB infection [55]. Both screening methods involve testing an individual for their immune response against Mtb antigens. The TST method involves injecting bacterial antigens intradermally and measuring the skin induration [21]. The IGRA method involves testing an individual's blood to check for IFN-g production against the Mtb antigen [21]. Those who have been vaccinated with the Bacillus Calmette-Guérin (BCG) vaccine, which is a prophylactic vaccine for TB, after their first year of life should be tested using the IGRA method rather than the TST method [45]. However, the IGRA method is less sensitive in children under the age of five, so the TST screening method is best suited for children under five years of age [45]. It was found that in children aged six months to five years, there was a difference found that shows the TST method is more sensitive than IGRA for detecting TB infection, especially in those with possible LTBI [56]. However, as there are mixed results with the TST and IGRA methods, it is recommended to test children under the age of five with both methods, especially for detecting LTBI [56].

## 4. Pediatric Immunology and Increased Susceptibility

A recent Stanford-led study that data-tracked 137,647 children from researchers around the world, starting from when a family member or close contact was diagnosed with active TB, showed that the risk of developing active disease was 19% in children as old as 5 and 8–12% in children ages 5–19, which is higher than historical estimates [57]. The conclusion was that children exposed to TB are at higher risk of active infection than was previously thought. In addition to the COVID-19 pandemic stay-at-home measures and the plummet in TB diagnoses, household transmission to children is a threat that cannot be overlooked.

Children under the age of five are more at risk for developing TB infection and even disseminated TB, which is when the mycobacterium has spread to other parts of the body other than the lungs. Children who have pre-existing conditions such as low body mass index, HIV, or any other kind of immunosuppressive disease have a much higher likelihood of contracting TB infection [58]. A study found that most children that have TB manifest as extrapulmonary TB, which means it has transferred to organs other than the lungs, indicating greater risk of a more severe disease course once infected [59]. From an immunological standpoint, children have fewer APCs in their bodies and therefore the pathogen can invade easier with less of an immune response [21].

In addition, within the first year of life, infants do not have fully mature Toll-like receptors to allow for phagocytosis of the pathogen, so this could also impair children's immune responses to mycobacterium [60]. In a study conducted by Carreto-Binaghi, it was found that some pediatric patients infected with tuberculosis had a lessened innate immune response due to low levels of IL-8, which prevented them from clearing the infection out of their system with their first line of defense [61]. A summary of immunodeficiencies in infants are shown in Figure 1. Young children and infants have an immature innate immune system, which allows the pathogen to pass through this barrier and enter the host cells. Pediatric TB cases are termed paucibacillary, which means there are lower numbers of the pathogen in circulation than adults [62]. As pediatric tuberculosis elicits different cytokines and T cells, as well as manifests in different organs, and has fewer mycobacteria

in the system when compared to adults, this should be kept in mind when managing pediatric patients. Due to their immune system being weaker at eliciting the most beneficial innate host defense mechanisms, pediatric patients are more susceptible to Mtb and more detrimental prognoses [61].

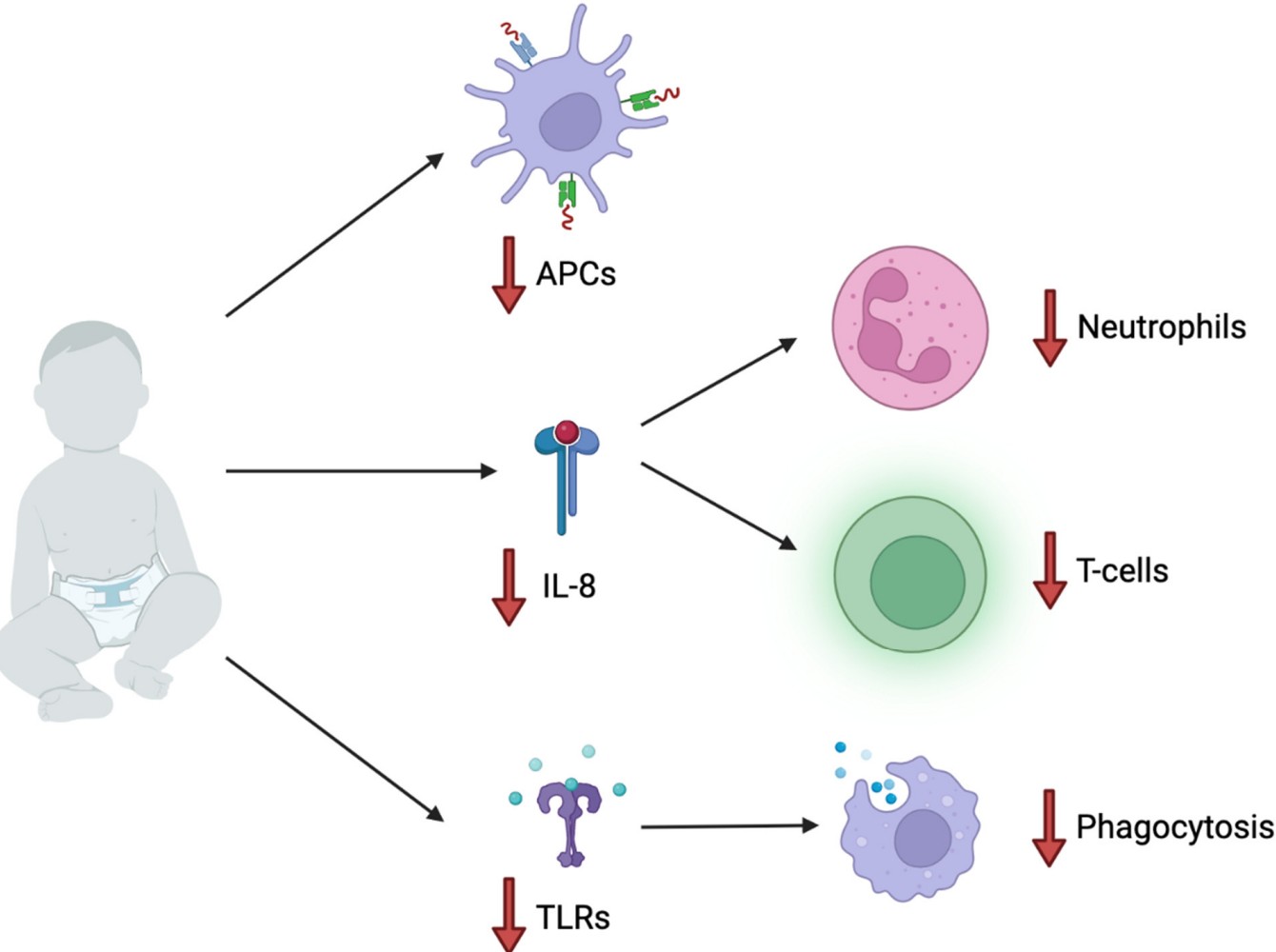

**Figure 1.** Infants are more susceptible to acquiring TB due to an immature immune system, including reduced antigen-presenting cells (APCs), reduced IL-8 chemokine production leading to impaired chemotaxis of neutrophils and T-cells, reduced toll-like receptors (TLRs), and, therefore, decreased phagocytosis for fighting infection.

## 5. SARS-CoV-2 Co-Infection in Children

With the onset of the COVID-19 pandemic caused by the pathogen SARS-CoV-2 at the beginning of 2020, infectious disease development has seen major impacts. In this section, we take a look at how co-infection with *M. tuberculosis* and SARS-CoV-2 can change pathogenesis in an individual. Both tuberculosis and COVID-19 are diseases that cause respiratory symptoms. In a normally healthy individual, these diseases take a toll on the host's immune system. When they present together, it can increase the chances of causing death due to the cytokine storm, multi-organ dysfunction, overhaul of the immune system, and coagulation activation [63,64]. With co-infection, the prognosis may be worse due to a higher burden on the immune system and could lead to more fatalities [65]. Both *M. tuberculosis* and SARS-CoV-2 infect and reproduce in the ciliated epithelial type 2 pneumocytes, as well as alveolar macrophages [66]. As these pathogens are targeting the same type of cells, the immune system can be exacerbated due to overworking to fight off co-infection with both pathogens. In *M. tuberculosis* infection, the cytokine

storm can also be triggered from an immune response caused by alveolar macrophages, monocytes, and dendritic cells [67]. These cells will release pro-inflammatory cytokines that lead to the cytokine storm. This, paired with co-infection of COVID-19 and the cytokine storm that occurs due to COVID-19, can exacerbate the host's immune system and lead to bad outcomes.

It has also been shown that both infection with SARS-CoV-2 and *M. tuberculosis* can lead to T lymphocyte dysregulation [68]. In COVID-19 infection, it is thought that the virus directly infects lymphocytes, which leads to the destruction of lymphocytes and a low count [69]. As stated in a previous section, the immune system utilizes T lymphocytes and CD4+ cells to fight off the *M. tuberculosis* infection. If there is a decreased level of T lymphocytes, there could be a high fatality rate and immunosuppression due to the inability to fight off SARS-CoV-2 and *M. tuberculosis*, which both implicate T lymphocytes in their pathogenicity.

Recent research has emerged suggesting that the BCG vaccine, which is used to build innate immunity against *M. Tuberculosis*, can also provide benefits to boost the immune system through increasing pro-inflammatory cytokines that include TNFa, IL1-B, IL6, IFN gamma, alveolar macrophages, and T cells [70]. With this increased immunity, it has shown that this BCG vaccine can provide benefits to improve the symptomology of patients that are also co-infected with COVID-19. It was found that those who were infected with SARS-CoV-2 and vaccinated with the BCG vaccine experienced a lessened viral load, viral replication inhibition, and reduced pro-inflammatory cytokine response that leads to better outcomes for COVID-19 patients [71]. There needs to be more research and clinical trials performed to understand the exact mechanisms of why this BCG vaccine provides a stronger immune response to fight co-infection with COVID-19.

Most recently, researchers from Massachusetts General Hospital published results from a "randomized, double-blinded, placebo controlled trial evaluating the safety and efficacy of a multi-dose BCG vaccine for the prevention of COVID-19 and other infectious disease in a COVID-19-unvaccinated, at-risk-community-based cohort" with type 1 diabetes. In 144 subjects, results showed a 92% efficacy for the BCG vaccine with 12.5% of placebo-treated subjects compared to 1% of BCG-treated participants having confirmed COVID-19 infection by the end of the study [72].

In a 2021 study in India by Mathur et al., 327 pediatric COVID-19 patients were admitted, of which 17 patients had concomitant TB. These patients had a more severe disease course with lower SpO$_2$ on arrival, higher mechanical ventilation rates, longer length of hospital stays, and worse outcomes. Considering that TB is preventable and treatable, the conclusion of their study pointed out once again to the importance of screening, diagnosing, and treating for pediatric tuberculosis, especially during the pandemic to prevent worse outcomes with co-infection [73].

## 6. Childhood Tuberculosis: Risk Factors

Economic status, sociocultural factors, behavioral patterns, and environmental components are a few factors that affect the degree and risk of human exposure to Mtb, and have all been impacted by COVID-19 [74]. Stay-at-home orders have also affected risk factors, leading to possible increased household transmissions during the pandemic.

Age is indicative of poor TB progression, with the greatest risk in infants as well as substantial, increasing risk in adolescents. In contrast, children ages 5–10 demonstrate decreased risk [75–77]. Furthermore, poverty is positively associated with increased risk of exposure to and acquiring TB. This can be attributed to insufficient immunization levels, poor quality of life, lack of education, lack of awareness regarding the pathogenesis of disease, overcrowding in spaces such as homes and schools, poor housing conditions, and lack of proper ventilation, which cumulatively increases the probability of developing poor outcomes [74,78–80]. Furthermore, urban locations where low socioeconomic and nonwhite ethnic groups reside have a higher incidence of TB cases [51,56]. This may be due to overcrowding in working, public, and living spaces due to increased population density;

this increase in proximity lends the opportunity for the spread of disease to vulnerable populations, such as children [81,82].

Additionally, household air pollution is implicated in increased TB risk. Children exposed to secondhand smoke, such as from an adult in the household, are at an increased risk of becoming infected with TB, possibly exacerbated by COVID-19 stay-at-home measures [76,83]. Although the exact mechanism warrants further investigation, possible reasons for the elevated risk may be due to the increased infectious nature of smokers who cough more, delayed diagnosis of TB because of chronic smoker's cough, and, hence, increased chances for transmission, or due to the detrimental effects of secondhand smoke on the respiratory immunity of children [84]. In addition, the use of solid fuels as energy sources for cooking increases the risk of TB. Cooking with solid fuels generates harmful substances such as carbon monoxide, NO, and polyaromatic hydrocarbons, with the last being a substance that can impede and weaken immune function, which adds to the already compromised immune system of pediatric patients, as discussed previously. Here, a lack of proper ventilation is a strong association factor that exacerbates the relationship between household air pollution and TB [84–86].

Furthermore, malnourishment and inappropriate low weight are associated with an elevated risk and burden of TB in children. Conversely, individuals who are overweight have a lower risk of TB. It is known that nutritional deficits are inherently linked to compromised immunity. Missing essential vitamins and minerals, such as vitamins A, C, D, and E or zinc, may impact immune function, which can lead to a decreased ability to fight off pathogens. Severe acute malnutrition (SAM) is linked to an increased incidence of infections such as TB, and its immunosuppressive effects facilitate poor outcomes in children and progression to active disease. Infection can exacerbate malnutrition, which, in turn, contributes to a further negative disease state [87–90].

Immunodeficiencies in children predispose them to infection by pathogens such as Mtb as well as the use of immunosuppressive regimens and drugs including but not limited to chemotherapy and corticosteroid therapy. Children who are positive for the human immunodeficiency virus (HIV) are at greater risk of TB infection [91,92]. Similarly, adults who are HIV-positive are at an increased risk of co-infection with TB as a result of comprised immune systems; thus, children living with adults who are HIV-positive have higher rates of exposure to TB infection [93–96]. A summary of different risk factors for childhood TB are shown in Figure 2.

Additionally, COVID-19 has placed heavy burdens on healthcare systems worldwide, leading to fewer resources to address TB as more are allocated to address COVID-19 and less vaccinations as fears of contagions and lockdowns spread [97,98]. These COVID-19-related factors have exacerbated TB disease risk. With lockdowns and more time spent at home, there is increased transmission of TB within the household [99]. A newly developed model analysis on the impact of COVID-19 on TB has demonstrated increases of 1.65 million TB cases and 438,000 TB deaths in India over the next 5 years due to a 3-month COVID-19 lockdown [100].

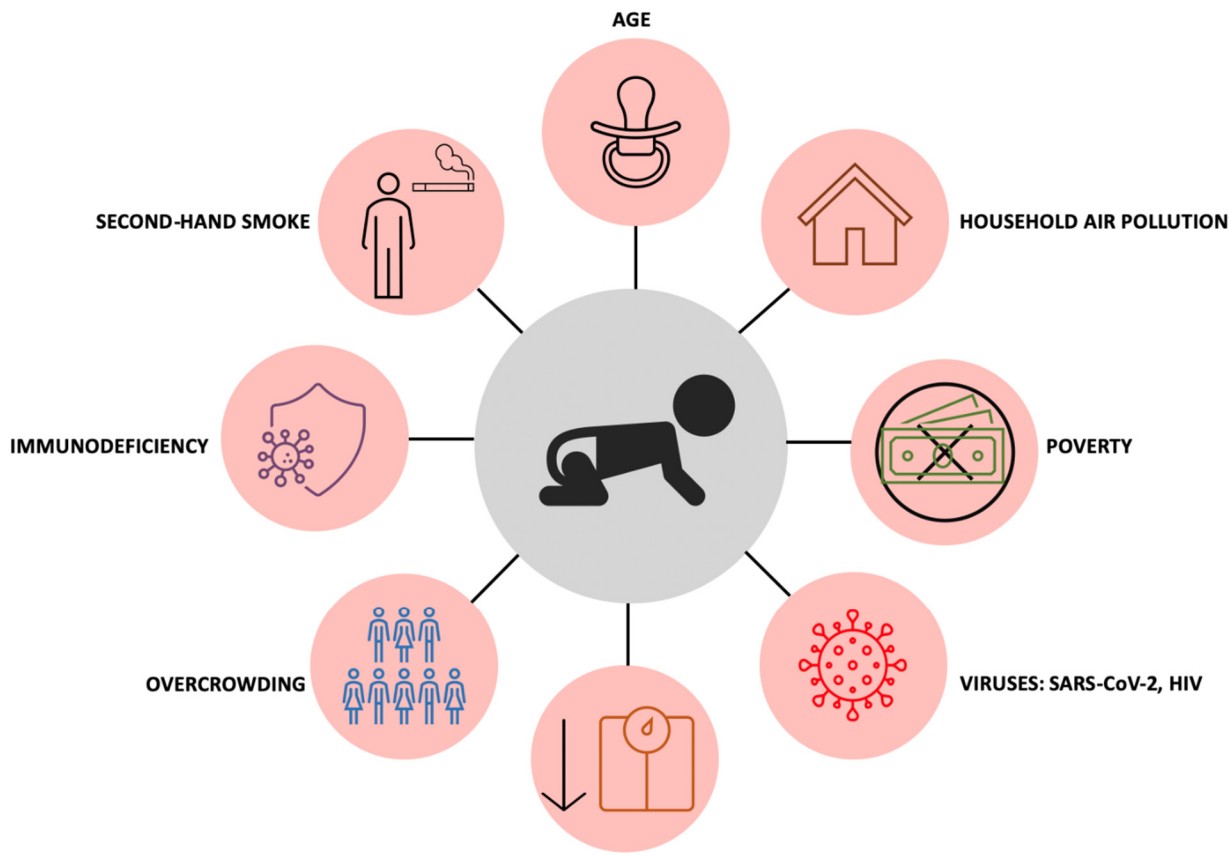

**Figure 2.** The likelihood of acquiring childhood TB is impacted by age, where infants and children below 5 years have the greatest risk, household air pollution, poverty, SARS-CoV-2 and HIV infection, low weight and malnutrition, overcrowding, immunodeficiency, and second-hand smoke exposure.

## 7. Current Screening and Diagnosis Guidelines

After reviewing the epidemiological trends of pediatric TB cases globally before and after the pandemic, as well as how children at certain ages are more immunologically susceptible to TB infection, including various socioeconomic risk factors involved in TB being one of the leading causes of pediatric mortality worldwide, we now provide an overview of current screening and diagnosis guidelines.

In the United States, it is currently not recommended for children to undergo routine TB screening. Ever since 1996, the American Academy of Pediatrics (AAP) has recommended targeted tuberculin skin testing (TST) of children while discouraging routine TST without risk factors, which is also consistent with most recent guidelines [101,102]. Reasons for this included false-positive cases in low-risk patients leading to unnecessary exposure to radiation through follow-up chest X-rays. Therefore, screening methods for pediatric TB cases in the US shifted from a universal testing approach to a universal screening and targeted testing approach in order to focus on more high-risk populations [103].

With the targeted testing approach, it is imperative for healthcare providers to administer the Pediatric TB Risk Assessment Questionnaire regularly at each annual physical examination. The standardized questionnaire screens for three main risk factors [104]. First, the patient identifies if they were born, have lived, or traveled to a country with an elevated TB rate for at least one month. If yes, then TB testing should occur at least 8 weeks after the child has returned from their travel. Secondly, further screening is also required if the child is immunosuppressed due to conditions such as HIV infection, organ transplant, or administration of immunosuppressive therapies including steroids and TNF-a antagonists. Finally, if the child has been in close contact with a TB-infected individual at any point,

then further screening is necessary. If any of the three risk factors are identified, then active TB should always be ruled out before initiating treatment for LTBI [105].

Using this same approach, it follows that both children and adults alike in high-risk countries, especially in Africa and South-East Asia, should be given routine TB testing. If this could be achieved in these particular areas of the world, the number of pediatric TB cases worldwide would significantly reduce. However, for obvious reasons, the feasibility of this is very low. In 2020, Imsanguan et al. reported the results of efforts in Thailand to increase contact tracing and follow-up testing in the province of Chiangrai, one of the poorest in the country with 143 TB deaths per 100,000 population, a mortality rate of 11.4%, and a HIV co-infection of 15.5% [105]. Patients were asked to give household and nonhousehold contacts invitations for free screening and chest X-rays with a stipend for transportation. Before the initiative, the coverage of contact investigation in under 5-year-olds at risk was only 33.2% (222 screened out of 668 contacts) over the period of 2011–2015. The results of the initiative yielded a 100% coverage of contact screening in under 5-year-olds with a TB detection rate of 21.4%. The results of this study once again show the importance of government initiatives in focusing on increasing contact tracing and screening, especially in impoverished and high-risk areas, especially now during the pandemic with climbing numbers of unreported cases and evidence of more severe initial presentation on diagnosis.

## 8. Impact of COVID-19 on Pediatric Tuberculosis Cases

Unfortunately, efforts to increase TB contact tracing and identify cases faced a large setback during COVID-19, with supposedly greater impacts on the pediatric population [6,106,107]. Before the global pandemic, TB disease diagnoses typically declined between 1 and 2% each year [108]. According to the CDC, in the U.S., reported TB cases decreased by 20% in 2020 and remained 13% lower in 2021 than TB diagnoses made prior to the COVID-19 pandemic [108]. Possible explanations to this include: (1) a reduction in the incidence of the disease due to increased respiratory precautions and social distancing, (2) delayed diagnoses due to widespread disruptions in healthcare during the pandemic, or (3) missed diagnoses from the overlap with COVID-19 symptoms [108]. It was found through case reports that some patients with TB were evaluated for COVID-19 but not tested for TB across multiple healthcare encounters [108]. As a result, the CDC launched the Think, Test, Treat TB campaign in March 2022 to continue to raise awareness for TB in the population and possibly to catch-up on any delayed diagnoses [109].

According to The New England Journal of Medicine, COVID-19 did in fact lead to increased missed TB diagnoses, and that the COVID-19 pandemic has had a disproportionately devastating effect on TB services with only 5.8 million cases reported in 2020 out of an estimated 10 million [2]. In 2020, there were roughly 1.5 million TB deaths worldwide, marking the first yearly increase in TB deaths since 2005 [2]. In addition, there was a 15% reduction in patients treated for drug-resistant TB, as well as a 21% decrease in individuals receiving preventative treatment [2].

Aznar et al. reported on the impact of COVID-19 on TB management in Spain. Across 2019 and 2020, it was found that patients diagnosed in 2020 showed more frequent bilateral lesions in chest X-rays than patients diagnosed in 2019 ($p = 0.004$), as well as a higher percentage of latent and active TB infection in children in households of patients diagnosed in 2020 compared to 2019 ($p = 0.001$) [6]. In another part of the world, Golandaj et al. from India reported a 24% decrease in reported pediatric TB cases during the overall lockdown period between March and May 2020 with an even greater 36% reduction post-lockdown from June to September of 2020 compared to the same months in 2019, which is concerning [110].

Overall, across several reports, there seems to be a consensus worldwide that COVID-19 has posed a major setback to TB-related targets, with trends expecting to worsen [2]. As data and reports continue to emerge around the world, we can anticipate a need for increased efforts to raise awareness for TB cases amidst COVID-19, in order to continue

adequate screening and treatment globally. Aznar et al.'s report in particular is especially concerning for rising cases of TB in children of patient households, reflecting increased household transmission during the pandemic, of which the exact causes remain unclear [6]. According to The Lancet, measures taken to control the spread of COVID-19 could have increased TB transmission in households because families were told to shelter in place together in addition to barriers to healthcare during the pandemic, reducing rapid diagnosis and prompt initiation of therapy [111]. It is also difficult to say exactly how the COVID-19 vaccination campaign has impacted prevention strategies for TB—for example, whether receiving mandatory COVID-19 shots would deter individuals from also receiving the BCG vaccine in certain parts of the world.

### 9. A Prevention Mechanism: BCG Vaccine

With downward trends on TB screening and diagnosis during the pandemic, it is even more important to raise awareness for the BCG vaccine during the pandemic. The Bacille Calmette–Guerin (BCG) vaccine, discovered by Calmette and Guerin in 1921, was developed from an attenuated Mycobacterium bovis strain and lends consistent efficacy against childhood TB when administered in a timely manner [112,113]. In areas with endemic TB, a single dose of BCG should be administered at birth or at the earliest opportunity if the vaccine is unavailable [114]. Routine administration of neonatal BCG demonstrates protective effects against miliary and meningeal TB where neonatal BCG was shown to reduce mortality in both TB-exposed and TB-unexposed children [41,115,116]. By altering the immune response, the BCG vaccine promotes a Mtb-targeted TH1 response [117,118]. IFN-$\gamma$, a cytokine critical in innate and adaptive immune responses and characteristic of a TH1 response, markedly increases from neonatal BCG administration to comparable levels to those of immune adults [118,119]. It was also shown that neonatal BCG is associated with increased CD4+, CD8+, TNF-$\alpha$, and IL-2 [120,121]. With the immunological memory and targeted TH1 response conferred by the BCG vaccine, children are better equipped to optimally respond to a Mtb infection [122]. Although ineffective against adult TB, there are various vaccine candidates undergoing development and trials with goals of improving BCG efficacy or providing an alternative vaccination source [112].

### 10. Prophylactic Treatment

Screening and diagnosis of pediatric TB is even more important during the pandemic, not just because children's immune systems are more at risk if exposed, as discussed earlier in length, but also because delayed diagnosis and treatment is a major factor in worse prognoses. Prompt management of LTBI with medications can prevent worsening progression to active TB and unwanted outcomes. Appropriate treatment regimens rely on knowledge of drug susceptibility/resistance, comorbidities, patient adherence, patient profile, and drug–drug interactions. Once a suitable treatment course is selected, children will undergo direct observed therapy (DOT) [123,124]. Although several LTBI treatments are available, much of their limited efficacy is compounded by poor adherence and unidentified or additional drug-resistant Mtb strains [125].

Isoniazid (INH) is an antibiotic with 2 treatment course options: a 6-month or 9-month course. For maximal efficacy, the 9-month course is recommended; however, that is often times not completed in children, due to its long duration; there is an overall stronger adherence to a 6-month regimen [123,126]. An alternative treatment option is 4 months of rifampicin [40,123]. In a comparative study by Cruz et al., it was noted that children receiving 4 months of rifampicin were 8 times more likely to complete the full course of treatment than children receiving 9 months of INH [28]. Both antibiotic options when completed in its entirety had similar rates of efficacy and safety [40]. Another promising treatment option is a 3-month course of isoniazid plus rifapentine, or isoniazid and rifampicine in countries or regions where rifapentine is not available. [40,113]. This regimen is associated with higher completion rates and few adverse effects in children [126,127]. With consumption of 2 medications, there are concerns of hepatotoxic effects; however, in a study by Villarino et al., it

was reported that no children developed hepatotoxicity [124,126]. This could potentially be attributed to the lower medication exposure duration [126]. Additionally, this treatment option is as efficacious as 9 months of INH [40,124]. Its shorter course might encourage greater adherence, prompt more treatment starts, and consequently better control TB in children [124].

## 11. Conclusions

With great setbacks on tuberculosis screening and diagnosis globally, but particularly in impoverished areas in Asia, and as the world gradually recovers from the devastation of the COVID-19 pandemic, there is a need for re-focused efforts especially from both healthcare providers and government organizations to improve TB prevention, screening, and diagnosis numbers to what it was pre-pandemic. This includes continuing to work for BCG vaccine equity. Before we can reach the World Health Organization's goals for the End Tuberculosis Strategy, we first need to recover from the aftermath of the pandemic.

As more data are revealed, pediatric TB has emerged to be a hidden threat behind the effects of COVID-19 as recent data showed how exposed pediatric patients are more susceptible to the disease than we previously thought [57], with signs of increased household transmissions from stay-at-home measures due to various risk factors such as tobacco smoking [6], as well as worse prognoses from co-infection [73], despite TB being a preventable and treatable disease. Ryckman et al. in their review article on "Ending tuberculosis in a post-COVID-19 world" suggested that a person-centered, equity-oriented response focusing on targeting the widened health disparities as a result of the pandemic is key to our approach moving forward [128]. As resources and attention around the world gradually come back to other infectious diseases such as TB, we can hope for current downward trends to stabilize back to the pre-pandemic era.

**Author Contributions:** P.R., P.A., M.N., V.Z. and V.V. contributed to drafting this review. P.R. and V.V. conceived the framework, provided guidance and assistance, and made edits to the draft. All authors have read and agreed to the published version of the manuscript.

**Funding:** This research received no external funding.

**Institutional Review Board Statement:** Not applicable.

**Informed Consent Statement:** Not applicable.

**Data Availability Statement:** Not applicable.

**Conflicts of Interest:** The authors declare no conflict of interest.

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
