# Peer review of "Review of Pediatric Tuberculosis in the Aftermath of COVID-19"

_clinpract, doi:10.3390/clinpract12050077_

Round 1
Reviewer 1 Report
The manuscript under review is an interesting review on the impact after the spreading of SARS-CoV-2 and COVID-19 pandemic on screening, diagnosis, and treatment strategies of pediatric tuberculosis. The paper is well-written and explicative, I have only a few minor comments:
- The Authors wrote in the whole manuscript “et. al”, however it should be “et al.” (it is from latin “et alii”).
- In Figure 2, “viruses: COVID-19, HIV” should be “viruses: SARS-CoV-2, HIV”.
- I suggest changing the title “introduction/Epidemiology” in just “Introduction”, as well as the title “Conclusion & Discussion” in “Conclusion”.
- In Authors opinion, does the COVID-19 vaccination campaign affect the preventing strategies for TB? If yes, add a few lines to explain.
- I suggest implementing the reference list with the following papers: 10.1186/s12941-020-00363-1; 10.3390/healthcare10020386; 10.1183/13993003.02105-2020; 10.3390/medicina58020144; 10.5588/ijtld.20.0417
Author Response
We appreciate your constructive comments and positive feedback. The manuscript was revised based on your recommendations. We look forward to publishing this important work. Many thanks.
REVIEWER#1
Review comments
The manuscript under review is an interesting review on the impact after the spreading of SARS-CoV-2 and COVID-19 pandemic on screening, diagnosis, and treatment strategies of pediatric tuberculosis. The paper is well-written and explicative, I have only a few minor comments:
- The Authors wrote in the whole manuscript “et. al”, however it should be “et al.” (it is from latin “et alii”).
Addressed
- In Figure 2, “viruses: COVID-19, HIV” should be “viruses: SARS-CoV-2, HIV”.
Addressed
- I suggest changing the title “introduction/Epidemiology” in just “Introduction”, as well as the title “Conclusion & Discussion” in “Conclusion”.
Addressed
- In Authors opinion, does the COVID-19 vaccination campaign affect the preventing strategies for TB? If yes, add a few lines to explain.
Lines added to explain how COVID-19 vaccination may impact TB prevention strategies
- I suggest implementing the reference list with the following papers: 10.1186/s12941-020-00363-1; 10.3390/healthcare10020386; 10.1183/13993003.02105-2020; 10.3390/medicina58020144; 10.5588/ijtld.20.0417
References added

Reviewer 2 Report
This is an excellent review of the problems of tuberculosis (TB) in children, stressing the impact of COVID-19 on epidemiology, screening and outcome of pediatric TB since 2020.
The paper is very comprehensive, but sometimes confuse as some informations (for instance the immunology of host defense against M. tbc) are repeated.
Furthermore, there are some confusion between TB infection (which is, by definition, a state of asymptomatic immune reaction) and disease (for instance line 163). TB infection and disease are not two distinct entities or "two possible manifestations" but two ends of a spectrum, all TB starting with infection (see for instance a good description in Esmail H, et al. 2014 The ongoing challenge of latent tuberculosis. Phil.Trans. R. Soc. B 369: 20130437. http://dx.doi.org/10.1098/rstb.2013.0437).
Minor comments:
1. Line 55: I suggest to add that the diagnosis of TB is made in children with symptoms, not in healthy and asymptomatic children
2. Line 153. The sentence gives the impression that TB is a disease transmitted from animals or from the soil to humans. This may have been true at the beginning of history but currently TB is mainly a disease transmitted between human beings.
3. Line 201. T cells kill the bacteria, not the disease
4. Line 236. If a child has documented LTBI (meaning that there is some proof if infection), testing is unnecessary. Testing is necessary in children where infection is suspected or possible because of exposure or symptoms
5. Line 256: "TLR" please spell in full in the text
6. Line 267: what is the difference in treatment of TB in children vs adults? Not in the WHO and Union Guidelines. Please revise or give evidence
7. Line 307: the evidence that BCG offers protection against COVID is very weak. Please provide evidence (and not only hypothesis)
8. Line 354: Unclear. Do patients with overweight have lower or higher risk of TB (as diabetics, who are usually overweight, have a higher risk
9. Line 390. Does the AAP still recommends a "universal screening" for TB in children? The 1996 Guidelines are outdated, please provide an updated reference
10. Line 506. The latest WHO recommendations include a 3-month course of rifampicine and isoniazid (in countries or regions where rifapentine is not available). See ref 93
11. Please check the references. There is no author "Organization, W.H." or "Geneva". This is a common misspelling linked to the use of EndNote if the comma "," is not placed at the correct place during entering the references
Author Response
We appreciate all your constructive comments and positive feedback. The manuscript was revised based on your recommendations. We look forward to publishing this important work. Many thanks.
REVIEWER#2
Review comments
This is an excellent review of the problems of tuberculosis (TB) in children, stressing the impact of COVID-19 on epidemiology, screening and outcome of pediatric TB since 2020.
The paper is very comprehensive, but sometimes confuse as some informations (for instance the immunology of host defense against M. tbc) are repeated.
Furthermore, there are some confusion between TB infection (which is, by definition, a state of asymptomatic immune reaction) and disease (for instance line 163). TB infection and disease are not two distinct entities or "two possible manifestations" but two ends of a spectrum, all TB starting with infection (see for instance a good description in Esmail H, et al. 2014 The ongoing challenge of latent tuberculosis. Phil.Trans. R. Soc. B 369: 20130437. http://dx.doi.org/10.1098/rstb.2013.0437).
This has now been corrected. Differentiated between infection vs. disease state.
Minor comments:
- Line 55: I suggest to add that the diagnosis of TB is made in children with symptoms, not in healthy and asymptomatic children
Comment added
- Line 153. The sentence gives the impression that TB is a disease transmitted from animals or from the soil to humans. This may have been true at the beginning of history but currently TB is mainly a disease transmitted between human beings.
Corrected
- Line 201. T cells kill the bacteria, not the disease
Corrected
- Line 236. If a child has documented LTBI (meaning that there is some proof if infection), testing is unnecessary. Testing is necessary in children where infection is suspected or possible because of exposure or symptoms
Corrected/clarified
- Line 256: "TLR" please spell in full in the text
Addressed
- Line 267: what is the difference in treatment of TB in children vs adults? Not in the WHO and Union Guidelines. Please revise or give evidence
Revised to reflect no difference in treatment
- Line 307: the evidence that BCG offers protection against COVID is very weak. Please provide evidence (and not only hypothesis)
Randomized controlled trial evidence added
- Line 354: Unclear. Do patients with overweight have loweror higherrisk of TB (as diabetics, who are usually overweight, have a higher risk
Clarified that overweight have lower risk, according to epidemiological evidence
- Line 390. Does the AAP still recommends a "universal screening" for TB in children? The 1996 Guidelines are outdated, please provide an updated reference
Updated reference provided
- Line 506. The latest WHO recommendations include a 3-month course of rifampicine and isoniazid (in countries or regions where rifapentine is not available). See ref 93
Added this information to our manuscript
- Please check the references. There is no author "Organization, W.H." or "Geneva". This is a common misspelling linked to the use of EndNote if the comma "," is not placed at the correct place during entering the references
References are now in order and links corrected
